# GRADIENTS AS FEATURES
# FOR DEEP REPRESENTATION LEARNING

**Fangzhou Mu,  Yingyu Liang**
Department of Computer Sciences
University of Wisconsin-Madison
`{fmu, yliang,}@cs.wisc.edu`

**Yin Li**
Departments of Biostatistics & Computer Sciences
University of Wisconsin-Madison
`yin.li@wisc.edu`

## ABSTRACT

We address the challenging problem of deep representation learning – the efficient adaption of a pre-trained deep network to different tasks. Specifically, we propose to explore gradient-based features. These features are gradients of the model parameters with respect to a task-specific loss given an input sample. Our key innovation is the design of a linear model that incorporates both gradient and activation of the pre-trained network. We demonstrate that our model provides a local linear approximation to an underlying deep model, and discuss important theoretical insights. Moreover, we present an efficient algorithm for the training and inference of our model without computing the actual gradients. Our method is evaluated across a number of representation-learning tasks on several datasets and using different network architectures. Strong results are obtained in all settings, and are well-aligned with our theoretical insights[1].

## 1 INTRODUCTION

Despite tremendous success of deep learning, training deep neural networks requires a massive amount of labeled data and computing resources. The recent development of representation learning holds great promises for improving data efficiency of training, and enables an easy adaption to different tasks using the same feature representation. These features can be learned via either unsupervised learning using deep generative models (Kingma & Welling, 2014; Dumoulin et al., 2016), or self-supervised learning with "pretext" tasks and pseudo-labels (Noroozi & Favaro, 2016; Zhang et al., 2016; Gidaris et al., 2018), or transfer learning from another large-scale dataset (Yosinski et al., 2014; Oquab et al., 2014; Girshick et al., 2014). After learning, the per-sample activation of the network is considered as generic features. By leveraging these features, simple classifiers such as linear models can be learned for different tasks. However, given sufficient amount of training data, the performance of representation-learning methods lags behind fully supervised deep models.

As a step towards bridging this gap, we propose to make use of gradient-based features from a pre-trained network, i.e., gradients of the model parameters with respect to a task-specific loss given an input sample. Our key intuition is that these per-sample gradients contain task-relevant discriminative information. More importantly, we design a novel linear model that accounts for both gradient- and activation-based features. The design of our linear model stems from the recent advances in the theoretical analysis of deep models. Specifically, our gradient-based features are inspired by the neural tangent kernel (NTK) (Jacot et al., 2018; Lee et al., 2019; Arora et al., 2019b), and adapt NTK in the setting of finite-width networks. Therefore, our model provides a local approximation of fine-tuning an underlying deep model, and the accuracy of the approximation is controlled by the semantic gap between the representation-learning and the target tasks. Finally, the structure of the gradient features and the linear model allows us to derive an efficient and scalable algorithm for training and inference.

To evaluate our method, we focus on visual representation learning in this paper, although our model can be easily modified for natural language or speech data. To this end, we consider a number of learning tasks in vision, including unsupervised, self-supervised and transfer learning. Our method

---

[1]Project webpage at `http://pages.cs.wisc.edu/~fmu/gradfeat20`

is evaluated across tasks, datasets and network architectures and compared against a set of baseline methods. We observe empirically that our model with the gradient features outperforms the traditional activation-based logistic regressor by a significant margin in all settings. Moreover, our model compares favorably against fine-tuning of network parameters.

Our main contributions are thus summarized as follows.

- We propose a novel representation-learning method. At the core of our method lies in a linear model that builds on gradients of model parameters as the feature representation.

- From a theoretical perspective, we claim that our linear model provides a local approximation of fine-tuning an underlying deep model. From a practical perspective, we devise an efficient and scalable algorithm for the training and inference of our method.

- We demonstrate strong results of our method across various representation-learning tasks, different network architectures and several datasets. Furthermore, these empirical results are well-aligned with our theoretical insight.

## 2 RELATED WORK

**Representation Learning**. Learning good representation of data without expensive supervision remains a challenging problem. Representation learning using deep models has been recently explored. For example, different types of deep latent variable models (Kingma & Welling, 2014; Higgins et al., 2017; Berthelot et al., 2019; Dumoulin et al., 2016; Donahue et al., 2016; Dinh et al., 2017; Kingma & Dhariwal, 2018; Grathwohl et al., 2019) were considered for representation learning. These models were designed to fit to the distribution of data, yet their intermediate responses were found useful for discriminative tasks. Another example is self-supervised learning. This paradigm seeks to learn from a discriminative pretext task whose supervision comes almost for free. These pretext tasks for images include predicting rotation angles (Gidaris et al., 2018), solving jigsaw puzzles (Noroozi & Favaro, 2016) and colorizing grayscale images (Zhang et al., 2016). Finally, the idea of transfer learning hinges on the assumption that features learned from a large and generic dataset can be shared across closely related tasks and datasets (Girshick et al., 2014; Sharif Razavian et al., 2014; Oquab et al., 2014). The most successful models for transfer learning so far are those pre-trained on the ImageNet classification task (Yosinski et al., 2014).

As opposed to proposing new representation-learning tasks, our work studies how to get the most out of the existing tasks. Hence, our method is broadly applicable – it offers a generic framework that can be readily combined with any representation-learning paradigm.

**Gradient of Deep Networks**. Our method makes use of the Jacobian matrix of a deep network as feature representation for a downstream task. Gradient information is traditionally employed for visualizing and interpreting convolutional networks (Simonyan et al., 2013), and more recently for generating adversarial samples (Szegedy et al., 2013), crafting defense strategies (Goodfellow et al., 2015), facilitating knowledge distillation (Sinha et al., 2018; Srinivas & Fleuret, 2018), and boosting multi-task and meta learning (Sinha et al., 2018; Achille et al., 2019).

Our work draws inspiration from Fisher vectors (FVs) (Jaakkola & Haussler, 1999) – gradient-based features from a probabilistic model (e.g., mixture of Gaussians). The FV pipeline has demonstrated its success for visual recognition based on hand-crafted local descriptors (Perronnin & Dance, 2007). More recently, FVs have shown promising results with deep models, first as an ingredient of a hybrid system (Perronnin & Larlus, 2015), and then as task embeddings for meta-learning (Achille et al., 2019). Our method differs from the FV approaches in two folds. First, it is not built around a probabilistic model, hence has distinct theoretical motivation as we describe later. Second, our method enjoys *exact* gradient computation with respect to network parameters and allows scalable training, whereas Achille et al. (2019) employs heuristics in their method to aggressively approximate the computation of FVs.

Another line of relevant research is from Zinkevich et al. (2017). They also explored similar gradient features of deep networks, which they call holographic features, by drawing inspiration from generalized linear models (GLM). They showed that a linear model trained on the gradient features can faithfully recover the output of the original network for the same task, and also showed some other desired properties of gradient features. In contrast, our work is motivated by the NTK theory,

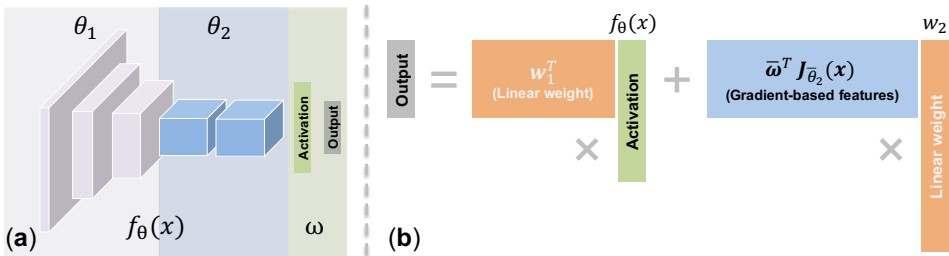

Figure 1: **(a)** An illustration of our parameterization. We consider a deep network $F(\boldsymbol{x}; \theta, \boldsymbol{\omega}) \triangleq \boldsymbol{\omega}^T f_\theta(\boldsymbol{x})$ that consists of a ConvNet $f_\theta$ with its parameters $\theta \triangleq (\theta_1, \theta_2)$ and linear weights $\boldsymbol{\omega}$. **(b)** An overview of our proposed model. Our model takes the activation $f_\theta(\boldsymbol{x})$ and the gradient $J_{\bar{\theta}_2}(\boldsymbol{x})$ as input (see illustration in (a)), and learns linear weights $\boldsymbol{w}_1$ (matrix) and $w_2$ (vector) for prediction.

and we study the more challenging problem of *adapting* the features for a different task. It would be interesting to investigate the connection between GLM and NTK.

**Neural Tangent Kernel (NTK) for Wide Networks**. Jacot et al. (2018) established the connection between deep networks and kernel methods by introducing the neural tangent kernel (NTK). Lee et al. (2019) further showed that a network evolves as a linear model in the infinite width limit when trained on certain losses under gradient descent. Similar ideas have been used to analyze wide networks (see, e.g., Arora et al. (2019b;a); Li & Liang (2018); Allen-Zhu et al. (2019a); Du et al. (2019); Allen-Zhu et al. (2019b); Cao & Gu (2020); Mei et al. (2019)). Our method is, to our best knowledge, the first attempt to materialize the theory in the regime of practical networks. Instead of assuming random initialization of network parameters as all the prior works do, we for the first time empirically evaluate the implication of pre-training on the linear approximation theory.

## 3 GRADIENT-BASED FEATURES FOR REPRESENTATION LEARNING

We start by introducing the setting of representation learning using deep models. Consider a feed-forward deep neural network $F_{\theta, \boldsymbol{\omega}}(\boldsymbol{x}) \triangleq \boldsymbol{\omega}^T f_\theta(\boldsymbol{x})$ that consists of a backbone $f_\theta(\boldsymbol{x})$ with its vectorized parameters $\theta$ and a linear model defined by $\boldsymbol{\omega}$ (*italic* for vectors and **bold** for matrix). Specifically, $f_\theta$ encodes the input $\boldsymbol{x}$ into a vector representation $f_\theta(\boldsymbol{x}) \in \mathbb{R}^d$. $\boldsymbol{\omega} \in \mathbb{R}^{d \times c}$ are thus linear weights that map a feature vector into $c$ output dimensions. For this work, we focus on convolutional networks (ConvNets) for classification tasks. With trivial modifications, our method can easily extend beyond ConvNets and classification, e.g., for a recurrent network as the backbone and/or for a regression task.

Following the setting of representation learning, we assume that a *pre-trained* $f_{\bar{\theta}}$ is given with $\bar{\theta}$ as the learned parameters. The term representation learning refers to a set of learning methods that do not make use of discriminative signals from the task of interest. For example, $f$ can be the encoder of a deep generative model (Kingma & Welling, 2014; Dumoulin et al., 2016; Donahue et al., 2016), or a ConvNet learned by using proxy tasks (self-supervised learning) (Goyal et al., 2019; Kolesnikov et al., 2019) or from another large-scale labeled dataset such as ImageNet (Deng et al., 2009). Given a target task, it is common practice to regard $f_{\bar{\theta}}$ as a fixed feature extractor (activation-based features) and train a set of linear weights, given by

$$g_{\bar{\boldsymbol{\omega}}}(\boldsymbol{x}) = \bar{\boldsymbol{\omega}}^T f_{\bar{\theta}}(\boldsymbol{x}), \tag{1}$$

We omit the bias term for clarity. Note that $\bar{\boldsymbol{\omega}}$ and $\bar{\theta}$ are instantiations of $\boldsymbol{\omega}$ and $\theta$, where $\bar{\boldsymbol{\omega}}$ is the solution of the linear model and $\bar{\theta}$ is given by representation learning. Based on this setup, we now describe our method, discuss the theoretical implications and present an efficient training scheme.

### 3.1 METHOD OUTLINE

Our method assumes a partition of $\theta \triangleq (\theta_1, \theta_2)$, where $\theta_1$ and $\theta_2$ parameterize the bottom and top layers of the ConvNet $f$ (see Figure 1(a) for an illustration). Importantly, we propose to use *gradient*-based features $\nabla_{\bar{\theta}_2} F_{\bar{\theta}, \bar{\boldsymbol{\omega}}}(\boldsymbol{x}) = \bar{\boldsymbol{\omega}}^T J_{\bar{\theta}_2}(\boldsymbol{x})$ in addition to *activation*-based features $f_{\bar{\theta}}(\boldsymbol{x})$. Specifically, $\boldsymbol{J}_{\bar{\theta}_2}(\boldsymbol{x}) \in \mathbb{R}^{d \times |\theta_2|}$ is the Jacobian matrix of $f_{\bar{\theta}}$ with respect to the pre-trained parameter

$\bar{\theta}_2$ from the *top* layers of $f$. Given the features $(f_{\bar{\theta}}(\boldsymbol{x}), \bar{\boldsymbol{\omega}}^T \boldsymbol{J}_{\bar{\theta}_2}(\boldsymbol{x}))$ for $\boldsymbol{x}$, our linear model $\hat{g}$, hereby considered as a classifier for concreteness, takes the form

$$\hat{g}_{\boldsymbol{w}_1, w_2}(\boldsymbol{x}) = \boldsymbol{w}_1^T f_{\bar{\theta}}(\boldsymbol{x}) + \bar{\boldsymbol{\omega}}^T \boldsymbol{J}_{\bar{\theta}_2}(\boldsymbol{x}) w_2 = g_{\boldsymbol{w}_1}(\boldsymbol{x}) + \bar{\boldsymbol{\omega}}^T \boldsymbol{J}_{\bar{\theta}_2}(\boldsymbol{x}) w_2, \tag{2}$$

where $\boldsymbol{w}_1 \in \mathbb{R}^{d \times c}$ are linear classifiers initialized from $\bar{\boldsymbol{\omega}}$, $w_2 \in \mathbb{R}^{|\theta_2|}$ are shared linear weights for gradient features, and $|\theta_2|$ is the size of the parameter $\theta_2$. Both $\boldsymbol{w}_1$ and $w_2$ are our model parameters that need to be learned from a target task. An overview of the model is shown in Figure 1(b).

Our model subsumes the linear model in Eq. (1) as the first term, and includes a second term that is linear in the gradient-based features. We note that this extra linear term is different from standard linear classifiers as in Eq. (1). In this case, the gradient-based features form a matrix and the weight vector $w_2$ is multiplied to each row of the feature matrix. Therefore, $w_2$ is shared for all output dimensions. Similar to standard linear classifiers, the output of $\hat{g}$ is further normalized by the softmax function and trained with the cross-entropy loss using labeled data from the target dataset.

Conceptually, our method can be summarized into three steps.

- **Pre-train the ConvNet $f_{\bar{\theta}}$.** This is accomplished by substituting in any existing representation-learning algorithm. Examples include unsupervised, self-supervised and transfer learning.
- **Train linear classifiers $\bar{\omega}$ using $f_{\bar{\theta}}(\boldsymbol{x})$.** This is a standard step in representation learning by fitting a linear model using "generic features".
- **Learn the linear model $\hat{g}_{\boldsymbol{w}_1, w_2}(\boldsymbol{x})$.** A linear model in the form of Eq. (2) is learned using gradient- and activation-based features. Note that the features are obtained when $\theta = \bar{\theta}$ is kept fixed, hence our method *requires no extra tuning of the parameter $\bar{\theta}$*.

### 3.2 THEORETICAL INSIGHT

**The key insight is that our model provides a local linear approximation to $F_{\theta_2, \boldsymbol{\omega}}(\boldsymbol{x})$.** This approximation comes from Eq. (2) – the crux of our approach. Importantly, our linear model is mathematically well motivated – it can be interpreted as the first-order Taylor expansion of $F_{\theta, \boldsymbol{\omega}}$ with respect to its parameters $(\theta_2, \boldsymbol{\omega})$ around the point of $(\bar{\theta}_2, \bar{\boldsymbol{\omega}})$. More formally, we note that

$$\begin{aligned} F_{\theta, \boldsymbol{\omega}}(\boldsymbol{x}) &\approx \bar{\boldsymbol{\omega}}^T f_{\bar{\theta}}(\boldsymbol{x}) + \bar{\boldsymbol{\omega}}^T \boldsymbol{J}_{\bar{\theta}_2}(\boldsymbol{x})(\theta_2 - \bar{\theta}_2) + (\boldsymbol{\omega} - \bar{\boldsymbol{\omega}})^T f_{\bar{\theta}}(\boldsymbol{x}) \\ &= \boldsymbol{\omega}^T f_{\bar{\theta}}(\boldsymbol{x}) + \bar{\boldsymbol{\omega}}^T \boldsymbol{J}_{\bar{\theta}_2}(\boldsymbol{x})(\theta_2 - \bar{\theta}_2) \\ &= \hat{g}_{\boldsymbol{\omega}, \theta_2 - \bar{\theta}_2}(\boldsymbol{x}). \end{aligned} \tag{3}$$

With $\boldsymbol{\omega} = \boldsymbol{w}_1$ and $\theta_2 - \bar{\theta}_2 = w_2$, Eq. (2) provides a linear approximation of the deep model $F_{\theta_2, \boldsymbol{\omega}}(\boldsymbol{x})$ around the initialization $(\bar{\theta}_2, \bar{\boldsymbol{\omega}})$. Our key intuition is that given a sufficiently strong base network, training our model approximates fine-tuning $F_{\theta_2, \boldsymbol{\omega}}$.

The quality of the linear approximation can be theoretically analyzed via the recent neural tangent kernel approach (Jacot et al., 2018; Lee et al., 2019; Arora et al., 2019b) or some related ideas (Arora et al., 2019a; Li & Liang, 2018; Allen-Zhu et al., 2019a; Du et al., 2019; Allen-Zhu et al., 2019b; Cao & Gu, 2020; Mei et al., 2019) when the base network $F_{\theta, \boldsymbol{\omega}}$ is sufficiently wide and at random initialization. Unlike prior works, we apply the linear approximation on pre-trained networks of practical sizes. We argue that such an approximation is useful in practice for the following reasons:

- **The pre-trained network provides a strong starting point.** Thus, the pre-trained network parameter $\bar{\theta}$ is *close* to a good solution for the downstream task. The key to good linear approximation is that the network output is stable with respect to small changes in the network parameters. The pre-trained base network also has such stability properties, which are supported by empirical observations. For example, the pre-trained network has similar predictions for a significant fraction of data in the downstream task as a fine-tuned network.
- **The network width required for the linearization to hold decreases as data becomes more structured.** An assumption made in existing analysis is that the network is sufficiently or even infinitely wide compared to the size of the dataset, so that the approximation can hold for *any* dataset. We argue that this is not necessary in practice, since the practical datasets are *well-structured*, and theoretically it has been shown that as long as the trained network is sufficiently

wide compared to the effective complexity determined by the structure of the data, then the approximation can hold (Li & Liang, 2018; Allen-Zhu et al., 2019a). Our approach thus takes advantage of the bottom layers to reduce data complexity in the hope that linearization of the top (and often the widest) layers can be sufficiently accurate.

## 3.3 SCALABLE TRAINING

Moving beyond the theoretical aspects, a practical challenge of our method is the high cost of computing $\hat{g}$ during training and inference. A naïve approach requires evaluating and storing $\bar{\boldsymbol{\omega}}^T \boldsymbol{J}_{\bar{\theta}_2}(\boldsymbol{x})$ for all $\boldsymbol{x}$. This is computationally expensive and can become infeasible as the number of output dimensions $c$ and the parameter size $|\theta_2|$ grow. Inspired by Pearlmutter (1994), we design an efficient training and inference scheme for $\hat{g}$. Thanks to this scheme, the complexity of training our model using gradient features is of the same magnitude as training a linear classifier on network activation.

Central to our scalable approach is the inexpensive evaluation of the Jacobian-vector product (JVP) $\bar{\boldsymbol{\omega}}^T \boldsymbol{J}_{\bar{\theta}_2}(\boldsymbol{x}) w_2$, whose size is the same as $c$. First, we note that

$$F_{\bar{\theta}+r w_2, \bar{\omega}}(\boldsymbol{x}) = F_{\bar{\theta}, \bar{\omega}}(\boldsymbol{x}) + r\bar{\boldsymbol{\omega}}^T \boldsymbol{J}_{\bar{\theta}_2}(\boldsymbol{x}) w_2 + o(r^2) \tag{4}$$

by first-order Taylor expansion around a scalar $r = 0$. Rearrange and take the limit of $r$ to 0, we get

$$\bar{\boldsymbol{\omega}}^T \boldsymbol{J}_{\bar{\theta}_2}(\boldsymbol{x}) w_2 = \lim_{r \to 0} \frac{F_{\bar{\theta}+r w_2, \bar{\omega}}(\boldsymbol{x}) - F_{\bar{\theta}, \bar{\omega}}(\boldsymbol{x})}{r} = \frac{\partial}{\partial r}\Big|_{r=0} F_{\bar{\theta}+r w_2, \bar{\omega}}(\boldsymbol{x}), \tag{5}$$

which can be conveniently evaluated via forward-mode automatic differentiation.

More precisely, let us consider the basic building block of $f$ – convolutional layers. These layers are defined as a *linear* function $h(\boldsymbol{z}_c; \boldsymbol{w}_c, b_c) = \boldsymbol{w}_c^T \boldsymbol{z}_c + b_c$, where $\boldsymbol{w}_c$ and $b_c$, which are part of $\theta$, are the weight and bias respectively, and $\boldsymbol{z}_c$ is the layer input. We denote the counterparts of $\boldsymbol{w}_c$ and $b_c$ in $w_2$ as $\tilde{\boldsymbol{w}}_c$ and $\tilde{b}_c$, i.e., $\tilde{\boldsymbol{w}}_c$ and $\tilde{b}_c$ are the linear weights applied to $\boldsymbol{w}_c$ and $b_c$. It can be shown that

$$\frac{\partial h(\boldsymbol{z}_c; \boldsymbol{w}_c + r\tilde{\boldsymbol{w}}_c, b_c + r\tilde{b}_c)}{\partial r} = h(\boldsymbol{z}_c; \tilde{\boldsymbol{w}}_c, \tilde{b}_c) + h(\frac{\partial \boldsymbol{z}_c}{\partial r}; \boldsymbol{w}_c, 0), \tag{6}$$

where $\frac{\partial \boldsymbol{z}_c}{\partial r}$ is the JVP coming from the upstream layer.

When a nonlinearity is encountered, we have, using the ReLU function as an example,

$$\frac{\partial ReLU(\boldsymbol{z}_c)}{\partial r} = \frac{\partial \boldsymbol{z}_c}{\partial r} \odot \mathbf{1}_{\boldsymbol{z}_c \geq 0}, \tag{7}$$

where $\odot$ is the element-wise product, $\mathbf{1}$ is the element-wise indicator function and $\boldsymbol{z}_c$ is the layer input. Other nonlinearities as well as pooling layers can be handled in the same spirit. For batch normalization, we fold them into their preceding convolutions.

Importantly, Eqs. (6) and (7) provide an efficient approach to evaluating the desired JVP in Eq. (5) by successively evaluating a set of JVPs *on the fly*. This chain of evaluations starts with the seed $\frac{\partial z_0}{\partial r} = 0$, where $z_0$ is the output of the section of $f$ parameterized by $\theta_1$ and can be pre-computed. $\bar{\boldsymbol{\omega}}^T \boldsymbol{J}_{\bar{\theta}_2}(\boldsymbol{x}) w_2$ can be computed along with the standard forward propagation through $f$. Moreover, during the training of $\hat{g}$, its parameters $\boldsymbol{w}_1$ and $w_2$ can be learned via standard back-propagation. In summary, our approach only requires a *single* forward pass through the fixed $f$ for evaluating $\hat{g}$, and a *single* backward pass for updating the parameters $\boldsymbol{w}_1$ and $w_2$.

**Complexity Analysis.** We further discuss the complexity of our method in training and inference, and contrast our method to the fine-tuning of network parameters $\theta_2$. Our forward pass, as demonstrated by Eqs. (6) and (7), is a chain of linear operations intertwined by element-wise multiplications. The second term in Eq. (6) forms the "main stream" of computation, while the first term merges into the main stream at every layer of the ConvNet $f$. The same reasoning holds for the backward pass. Overall, our method requires twice as many linear operations as fine-tuning $\theta_2$ of the ConvNet. Note, however, that half of the linear operations by our method are slightly cheaper due to the absence of the bias term. Moreover, in the special case where $\theta_2$ only includes the topmost layer of $f$, our method carries out the same number of operations as fine-tuning since the second term in Eq. (6) can be dropped. For memory consumption, our method requires to store an additional "copy" (linear weights) of the model parameters compared to fine-tuning. As the size of $\theta_2$ is typically small, this minor increase of computing and memory cost puts our method on the same page as fine-tuning.

## 4 EXPERIMENTS

Our experimental results are organized into two parts. We first perform ablation studies to understand the representation power of the gradient features. Next, we evaluate our method on three representation-learning tasks: learning deep generative models, self-supervised learning using a pretext task, and transfer learning from ImageNet.

We report results on several datasets and using different base networks to demonstrate the strength of our method. In all cases, our method achieves significant better performance than the standard logistic regressor on network activation. In the most interesting scenario where the pre-training task is similar to the target task, our method achieves comparable or even better performance than fine-tuning. Before we present our results, we summarize our implementation details.

**Implementation Details**. We adopt the NTK parametrization (Jacot et al., 2018) for $\theta_2$ and fold batch normalization into their preceding convolutional layers prior to training our model. For the SVHN and CIFAR-10/100 experiments, We train the models for 80K iterations with initial learning rate 1e-3, halved every 20K iterations. For the VOC07 and COCO2014 experiments, we train the models for 50 epochs with initial learning rate 1e-3, halved every 20 epochs. All models are trained with the Adam optimizer (Kingma & Ba, 2015) with batch size 64, $\beta_1 = 0.5$, $\beta_2 = 0.999$ and weight decay 1e-6. In addition to Adam, we also use the SGD optimizer with weight decay 5e-5, momentum 0.9 and the same learning rate schedule for fine-tuning, and we report the better result between the two runs.

### 4.1 ABLATION STUDY

We conduct ablation studies to address the following two questions: Does pre-training encourage more powerful gradient features? What is the optimal size of the gradient features? We answer the first question by comparing gradient features derived from various combinations of random and pre-trained network parameters, and the second by varying the number of layers that contribute to the gradient features.

**Baselines**. We compare our linear model in Eq. (2), referred to as the **full** model in the sequel, against two baselines. The first baseline $g_{\boldsymbol{w_1}}(\boldsymbol{x})$, or the **activation** model, is the first term in the full model. It is a logistic regressor on network activation and is widely used for representation learning in practice. The second baseline $\bar{\boldsymbol{\omega}}^T \boldsymbol{J}_{\bar{\theta}_2}(\boldsymbol{x}) w_2$, or the **gradient** model, is the second term in the full model. It is linear in the gradient features and can serve as a linear classifier on its own. Moreover, we compare our method against fine-tuning in order to assess the validity of our theoretical insight.

**Settings**. For ablation study, we consider both unsupervised- and supervised-learning settings and report classification accuracy. In the *unsupervised setting* (Table 1), we transfer features learned from a deep generative model for image classification. Specifically, we train a BiGAN on CIFAR-10 (Krizhevsky et al., 2009) and use its encoder as the base network for classification of the same dataset. In the *supervised setting* (Table 2), we consider fine-tuning ImageNet pre-trained models on a different classification tasks. Concretely, we use the PyTorch (Paszke et al., 2017) distribution of ImageNet pre-trained ResNet18 (He et al., 2016) as the base network for VOC07 (Everingham et al., 2010) object classification. For both settings, activation features are *always* output from the pre-trained base network. In contrast, gradient features are evaluated with respect to all possible combinations of random and pre-trained network parameters. In this way, we ensure that variation in performance of the full model can be solely attributed to the gradient features.

**Results and Conclusions**. Our results are shown in Table 1 (unsupervised) and Table 2 (supervised). We summarize our main conclusions from the ablation study.

- **The discriminative power of the gradient features is a consequence of pre-training**. In particular, the full model's performance gain is not a consequence of an increase in parameter size. The full model supplied with random gradients is no better than the activation baseline.
- **Gradient from the topmost layer of a pre-trained network suffices to ensure a reasonable performance gain**. Further inflating the gradient features from bottom layers has diminishing returns, introducing little accuracy gain and sometimes hurting the performance. The results in Zinkevich et al. (2017) also suggest that gradient features from top layers are the most effective in terms of approximating the underlying network.

Table 1: **Ablation study in the unsupervised setting**. We train a BiGAN on CIFAR-10 and use its encoder as the base network for classification of the same dataset. All results are reported using classification accuracy. We obtain the pre-trained $\bar{\theta}_1$ and $\bar{\theta}_2$ from the pre-trained base network, and $\bar{\omega}$ by first training the activation baseline for the target task. The specifications of $\theta_1$, $\theta_2$ and $\omega$ only affects gradient evaluation. Activation features are always output from the pre-trained base network.

| | | | $\theta_2$: conv5 | | $\theta_2$: conv4-5 | | $\theta_2$: conv3-5 | |
|---|---|---|---|---|---|---|---|---|
| | | | grad | full | grad | full | grad | full |
| Random $\bar{\theta}_1$ | random | random $\omega$ | 23.09 | 62.83 | 26.30 | 62.83 | 27.52 | 62.85 |
| | $\theta_2$ | pre-trained $\bar{\omega}$ | 40.23 | 63.11 | 43.36 | 63.42 | 44.36 | 63.38 |
| | pre-trained | random $\omega$ | 32.50 | 62.98 | 33.20 | 63.10 | 35.13 | 63.10 |
| | $\bar{\theta}_2$ | pre-trained $\bar{\omega}$ | 44.07 | 63.99 | 45.87 | 64.21 | 48.88 | 64.51 |
| Pre-trained $\bar{\theta}_1$ | random | random $\omega$ | 53.39 | 63.74 | 56.24 | 64.30 | 57.20 | 64.60 |
| | $\theta_2$ | pre-trained $\bar{\omega}$ | 59.98 | 65.46 | 61.47 | 65.38 | 59.89 | 64.73 |
| | pre-trained | random $\omega$ | 70.28 | 69.84 | **71.08** | 70.10 | **71.60** | 70.64 |
| | $\bar{\theta}_2$ | pre-trained $\bar{\omega}$ | 70.14 | **70.51** | 70.89 | 70.78 | **71.55** | 71.37 |
| **activation** | | | 62.87 | | | | | |
| *fine-tuning* | | | *71.78* | | *73.18* | | *74.30* | |

Table 2: **Ablation study in the supervised setting**. We use the PyTorch distribution of ImageNet pre-trained ResNet18 as the base network for VOC07 object classification. All results are reported using mean average precision (mAP). Predictions are based on a single center crop at test time.

| | | | $\theta_2$: layer4 block2 | | $\theta_2$: layer4 block1-2 | |
|---|---|---|---|---|---|---|
| | | | grad | full | grad | full |
| Random $\theta_1$ | random | random $\omega$ | 15.63 | 82.84 | 17.96 | 82.85 |
| | $\theta_2$ | pre-trained $\bar{\omega}$ | 17.43 | 82.79 | 19.90 | 82.82 |
| | pre-trained | random $\omega$ | 18.15 | 82.85 | 20.15 | 82.21 |
| | $\bar{\theta}_2$ | pre-trained $\bar{\omega}$ | 20.78 | 82.76 | 23.57 | 82.73 |
| Pre-trained $\bar{\theta}_1$ | random | random $\omega$ | 62.35 | 82.84 | 69.34 | 82.82 |
| | $\theta_2$ | pre-trained $\bar{\omega}$ | 65.75 | 82.84 | 71.73 | 82.86 |
| | pre-trained | random $\omega$ | 80.74 | 83.15 | 80.30 | 82.97 |
| | $\bar{\theta}_2$ | pre-trained $\bar{\omega}$ | 83.05 | **83.50** | 83.12 | **83.40** |
| **activation** | | | 82.65 | | | |
| *fine-tuning* | | | *82.97* | | *82.50* | |

- **Our results improve as the dataset and base network grow in complexity**. Our method *always* outperforms the activation baseline by a significant margin when the gradient features are derived from *pre-trained* parameters. Moreover, our method consistently beats the gradient baseline and even fine-tuning on the more challenging VOC07 dataset when using a more complicated residual network.

**Remarks**. We obtain the pre-trained $\bar{\theta} = (\bar{\theta}_1, \bar{\theta}_2)$ from representation learning, and the pre-trained $\bar{\omega}$ by first training the standard logistic regressor for the target task. We follow the same training procedure in later experiments. According to the ablation study, a good heuristic for our method is to use the gradients from the *topmost* convolutional layer (or residual block) of a pre-trained network. A more principled strategy for selecting gradient-contributing layers is left for future work.

## 4.2 RESULTS ON REPRESENTATION LEARNING

We now present results on three different representation-learning tasks: unsupervised learning using generative modeling, self-supervised learning using pretext tasks and transfer learning from pre-trained models. For all experiments in this section, we contrast our proposed linear classifier (i.e., the full model in Eq. (2)) with the activation and gradient baselines as well as fine-tuning $\theta_2$ for the

Table 3: **Unsupervised Learning Results**. We consider two base networks, namely the encoder of a BiGAN and that of a VAE, trained on three datasets. Our target task is image classification on the same datasets. All results are reported using classification accuracy.

|  |  | SVHN | CIFAR-10 | CIFAR-100 |
|---|---|---|---|---|
| BiGAN ($\theta_2$: conv5) | **activation** | 82.04 | 62.87 | 34.30 |
|  | **gradient** | **89.91** | 70.14 | 38.99 |
|  | **full** | **89.96** | **70.51** | **39.37** |
|  | *fine-tuning* | *91.15* | *71.78* | *41.05* |
| VAE ($\theta_2$: conv8) | **activation** | 81.95 | 52.05 | 29.20 |
|  | **gradient** | **91.44** | **61.47** | **35.10** |
|  | **full** | 90.90 | 61.16 | 34.83 |
|  | *fine-tuning* | *93.61* | *65.16* | *37.86* |

Table 4: **Self-supervised and Transfer Learning Results**. We consider a ResNet50 pre-trained on the jigsaw pretext task, and a ResNet18 pre-trained on the ImageNet classification task. Our target task is VOC07 and COCO2014 object classification. All results are reported using mean average precision (mAP). Predictions are averaged over ten random crops at test time.

|  |  | VOC07 | COCO2014 |
|---|---|---|---|
| Self-supervised (Jigsaw) | **activation** | 57.83 | 42.10 |
|  | **gradient** | 57.73 | 41.02 |
|  | **full** | **61.70** | **45.62** |
|  | *fine-tuning* | *67.88* | *52.66* |
| Self-supervised (Colorization) | **activation** | 52.30 | n/a |
| Transfer (ImageNet) | **activation** | 83.59 | 62.98 |
|  | **gradient** | 84.63 | 66.33 |
|  | **full** | **84.95** | **66.45** |
|  | *fine-tuning* | *84.14* | *63.43* |

target task. Following our conclusion from the ablation study, we make use of gradient features only from the topmost layer (or residual block) of a pre-trained base network.

**Unsupervised Learning**. Similar to our ablation study, we consider unsupervised representation learning using deep generative models. We train both BiGAN and VAE, and use their encoders as the pre-trained ConvNet $f$. Our BiGAN and VAE models follow the architecture and training setup from Dumoulin et al. (2016) and Berthelot et al. (2019). We average-pool the ConvNet output for activation features, and obtain gradient features from the topmost convolutional layer. We train on the `train` split of CIFAR-10/100 and the `extra` split of SVHN, and report classification accuracy on their `test` splits.

**Unsupervised Learning Results**. Our results are summarized in Table 3. Our model consistently improves the activation baseline by over 10% on all three datasets. Moreover, we observe good agreement of performance between our model and fine-tuning. Finally, we notice that our model does not have significant advantage over the gradient baseline. We think this is because the datasets and base networks used here are so simple that the gradient features alone can be equally effective.

**Self-supervised Learning**. Moving beyond unsupervised learning, we also consider self-supervised setting, where the representation is learned via a pretext task and pseudo-labels. In this case, we consider a ResNet50 pre-trained on the jigsaw self-supervising task on ImageNet provided by Goyal et al. (2019). See Noroozi & Favaro (2016) for details on the jigsaw task. We average-pool the ConvNet output for activation features, and use gradient features from the last residual block. We train on the `trainval` split of VOC07 and the `train` split of COCO2014 for object classification, and report the mean average precision (mAP) scores on their respective `test` and `val` splits.

**Self-supervised Learning Results**. Our results are summarized in the top part of Table 4. We again observe a significant improvement of up to 4.5% over the two baselines on both datasets. Our method also outperforms a competitive self-supervised learning method (Colorization) (Zhang et al., 2016) using the activation from the same backbone network (ResNet50) and pre-trained on the same dataset (ImageNet). We note that our method still lags behind fine-tuning in this setting with a large gap (-6% on VOC07 and -7% on COCO2014).

**Transfer Learning**. Finally, we consider the most widely used transfer learning setting, where the representation is learned by pre-training on a large scale labeled dataset, e.g., ImageNet. We experiment with an ImageNet pre-trained ResNet18 from PyTorch. We follow the same setting as self-supervised learning and report results on VOC07 and COCO2014 datasets.

**Transfer Learning Results.** Our results are presented in the bottom part of Table 4. Not surprisingly, our results consistently outperforms the two baselines across both datasets. However, the gap between our method and gradient baseline is much smaller. To our surprise, our method outperforms fine-tuning on both datasets by a large margin. In comparison to fine-tuning, our method has a gain of +0.8% and +3.0% on VOC07 and the larger COCO2014 dataset.

## 4.3 Further Discussions

We provide further discussions of our results. Specifically, we analyze the performance gap between out method and fine-tuning under three different settings. We then compare our method with learning a linear classifier only using gradient features. Moreover, we contrast our method with the recent effort of learning infinitely wide networks using exact NTK (Arora et al., 2019b).

**The performance gap between our method and fine-tuning.** We argue that this gap is controlled by the degree of semantic overlap between the representation-learning and target tasks. The jigsaw pretext task, though proven effective for learning features, is still semantically distant to an image classification task, and hence results in the large performance gap we observe in the self-supervised setting. On the other hand, it is reasonable to expect classifiers for datasets with overlapping categories to share common semantics. This may explain why our method outperforms fine-tuning in the transfer-learning experiments, where the source and target datasets are conceptually close.

**Comparing our method with training a linear classifier on gradient features alone.** In general, both methods are considerably more powerful than the standard activation-based logistic regressor. In the unsupervised setting, the gradient model performs on par with, or even better than the full model, which consumes both activation and gradient features (see Table 3). This raises the question of whether network activations are redundant in the presence of gradient features. We argue that this is not necessarily the case. In fact, as we demonstrate in the self-supervised and transfer-learning experiments (see Table 4), combining activation and gradient features is a formula for delivering the best performance on challenging representation-learning tasks (jigsaw pretext task) and datasets (VOC07 and COCO2014). It would be interesting to further characterize the "representation bias" induced by the two types of features.

**Contrasting our method with learning infinitely wide networks from scratch.** Arora et al. (2019b) recently proposed exact NTK computation algorithms for learning infinitely wide ConvNets, and proved that such networks are equivalent to kernel regression using NTK. They obtained 66% accuracy on CIFAR-10 classification using a vanilla network and achieved an extra 11% boost after including customized network modules. Their approach, at its current stage, only supports full-batch learning and is limited to single output dimension. Our method, though not directly comparable with theirs, does not suffer from these limitations and delivers competitive results (e.g., we obtain 70.5% accuracy on CIFAR-10 using a four-layer BiGAN encoder as the base network, see Table 3). Hence, out work highlights another promising direction to capitalize on the insights generated by the theoretical analysis of NTK.

## 5 Conclusion

We presented a novel method for deep representation learning. Specifically, given a pre-trained deep network, we explored as features the per-sample gradients of the network parameters relative to a task-specific loss, and constructed a linear model that combines network activation with the gradient

features. We showed that our model can be very efficient in training and inference, and may be understood as a local linear approximation to an underlying deep model by an appeal to the neural-tangent-kernel (NTK) theory. We empirically demonstrated that the gradient features are highly discriminative for downstream tasks, and our method can significantly improve over the baseline method of representation learning across pre-training tasks, network architectures and datasets. We believe that our work offers a novel perspective to improving deep representation learning. Future research directions include using NTK as a distance metric for measuring sample similarity in low-shot classification and assessing model similarity in knowledge distillation.

**Acknowledgement:** This work was supported in part by FA9550-18-1-0166. The authors would also like to acknowledge the support provided by the University of Wisconsin-Madison Office of the Vice Chancellor for Research and Graduate Education with funding from the Wisconsin Alumni Research Foundation.

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
