# OpenReview forum: "Gradients as Features for Deep Representation Learning"
_ICLR.cc/2020/Conference — Accept (Poster)_

### Official Review · AnonReviewer2 · 2019-10-23
**Official Blind Review #2**

**Rating:** 6

**Review:**

Summary: This paper proposes to use the gradients of specific layers of convolutional networks as features in a linearized model for transfer learning and fast adaptation. The method is theoretically backed by an appeal to the recently proposed neural tangent kernel and seems like it could be practically useful.

Edit: Post rebuttal, I am somewhat satisfied by the authors' response and find their ablation study somewhat compelling hence am increasing my score to a weak accept. However, I'm not completely convinced that their choice of layers to use as gradient features is reasonable. If this paper does end up getting accepted, I'd very strongly advocate for being extremely clear as to
1) why the gradient features used are used,
2) a good choice of gradient features in practice
3) inclusion of all of the relevant ablation studies in the final version.

I tend to weakly reject this paper currently despite liking the simplicitly and timeliness of the approach. Specifically, I would like further discussion of the choice of the layers to use as gradient features and an ablation study on supervised trained networks.

Originality: I believe that this is one of the first papers to explicitly use the Taylor approximation (neural tangent kernel) in a transfer learning setting, making the approach timely and potentially practically useful.

Significance: The approach is a quite nice merging of theoretical insights with a neat practical implementation. Although the main methodological advance is a straightforward application of the thinking behind Jacobian vector products, the method is well described and ought to be practically useful. However, I have a bit of a concern as to its practical necessity in comparison to simple fine-tuning of the final softmax layer (referred to in Table 1 as \theta_2) on a new dataset.

Clarity: While the approach described in Section 3 is quite generic – theoretically, the method should simply consist of training the final layer of the neural network (w_1) and weights from the Jacobian features around the Taylor expansion. However, the experimental approaches suggest that different layers were used in each experiment – see e.g. “[we] … compute our gradient based features from one, two, or all of the top-3 conv layers” (Section 4.2 for the BiGAN architectures) versus “our gradient based features are from the last 2 conv layers” (Table 2 for the AlexNet architectures). Why was the entire network (modulo the last feed forward layer) not simply used for the Jacobian features?

Similarly, why was the entirety of the model (again modulo the last feedforward layer) not used for forwards model in the AlexNet experiments?

Based on the ablation study in Section 4.1, the authors find that “it suffices to set the very top layer as \theta_2 to enjoy a reasonably large performance gain.” When this is the case, it is a bit tough to distinguish the approach from standard final layer transferring approaches (Sharif Razavian et al, 2014) and indeed Bayesian final layer approaches that simply retrain an output layer (e.g. Perrone et al, 2018). Indeed this seems to be the case in Table 1, if that is so, then why not just re-train \theta_2 as well throughout the experiments as it will typically just be another stochastic convex optimization problem?

Could the authors quickly describe an implementation of the scalable Jacobian inner products in Section 3.3? It seems like outputs will have to be cached during the forwards passes, thereby requiring a somewhat significant amount of software engineering (and memory overhead) to be have to do this process for each new layer. Is this the correct understanding of how one would implement the procedure? A quick paste of PyTorch pseudo-code would be sufficient here.

Quality: The experiments seem to be relatively convincing – it seems exciting that a linear model + the network’s features itself can typically perform as well as fine-tuning and occasionally even better than fine-tuning itself.

However, I am a bit concerned by the fact that the ablation studies themselves only utilize models trained in an un-supervised fashion.
I’d instead like the authors to run an ablation study on supervised trained models (perhaps 8 layer conv nets or VGG16) in the same manner as in Table 1 and Section 4.1. Specifically, I’d like to see this done to see whether the gradients in fact are as interesting as features when they have been trained in a supervised fashion.

Similarly, I’d like to see the features themselves used as a linear classifier (no network forwards passed) in the same two ablation studies. That is, could the authors use w_1^T J w_2 as the features for their linear classifier. If they have already done so and I’ve missed that in the tables somehow, I apologize. This experiment should help to test out how _useful_ the features defined by the Jacobian matrix are in comparison to the network’s forward pass itself.

Minor Comments:

Introduction: “After learning, the the (sic) activation of the deep network are considered as generic features.” Not only is there a small typo, but you should either include a citation here or be more specific as to what “the activation” of the deep network is here.

Introduction: “And the accuracy of the rthe (sic) …” typo + please do not start sentences with and if at all possible.

Section 3.1 (beneath eq. 2): “liner” should be linear.

Table 3: “Self-supervise” should be “Self-supervised.”

References:

Perrone et al, Scalable Hyperparameter Transfer Learning, NeurIPS, 2018. http://papers.nips.cc/paper/7917-scalable-hyperparameter-transfer-learning.pdf

Sharif Razavian et al, CNN Features off-the-Shelf: an Astounding Baseline for Image Recognition, CVPRW, 2014. https://arxiv.org/abs/1403.6382



**Experience Assessment:**

I have read many papers in this area.

**Review Assessment: Checking Correctness Of Derivations And Theory:**

I assessed the sensibility of the derivations and theory.

**Review Assessment: Checking Correctness Of Experiments:**

I assessed the sensibility of the experiments.

**Review Assessment: Thoroughness In Paper Reading:**

I read the paper at least twice and used my best judgement in assessing the paper.

---

> ### Author Response · Authors · 2019-11-15
> **Response to Reviewer #2 Part I**
>
> Thanks for your very detailed comments and for your appreciation of the novelty and significance of our work. We clarify some of the technical details that might have created confusion in the review. We also discuss the choice of layers for computing gradient feature. More importantly, we include additional ablation study on supervised pre-training and demonstrate that our model consistently outperforms all baselines (including using gradient feature only). Finally, we present pseudo-code of our implementation. We hope that our response can address your concerns and kindly ask the reviewer to re-consider the rating.
>
> We present the key results in our response and put the full set of results temporarily in the appendix of the paper. These results will be merged into a future version of the paper. Please do not hesitate to ask if you have further questions.
>
> [General clarification]
>
> Before we address any specific concerns, we would like to clarify our approach using a toy example. Recall that our model, which can be intuitively understood as the linearization of a pre-trained network, contains two terms. The first term is the usual logistic regressor on the activation, and the second is linear in the per-sample Jacobian of the activation w.r.t. a set of network parameters, which we denote by $\theta_2$ in the paper. Suppose that a network $F(x)=g(f(x))$ has the architecture (conv1-conv2-conv3-conv4-fc). According to our notation, we refer to (conv1-conv2-conv3-conv4) as the ConvNet $f$ (parameterized by $\theta_1$ and $\theta_2$), and the last fc layer as $g$ (parameterized by $\omega$). When we say "gradient feature from the top layer of $f$", we mean the per-sample gradient w.r.t. the weight and bias of conv4, not the fully connected layer fc. Note, however, that our model still makes use of the gradient w.r.t. fc, which is exactly the activation from $f$ and hence is naturally merged into the logistic regressor term.
>
> Now we turn to the concerns raised by the reviewer.
>
> [On the choice of the layers for computing gradient feature]
>
> In the experiments, we choose to compute gradient feature from the topmost conv layers (or residual blocks), but not the entirety of the network for the following reasons.
>
> a) Width requirement. Empirical studies demonstrated that a network is well captured by its linearization only if it is sufficiently wide [1]. The most popular network architectures these days are almost always widest in the top.
>
> b) Diminishing gain. According to our ablation studies, the model using gradient from the topmost layer alone is already satisfying. Adding gradient from layers further down the hierarchy results in very little performance gain. This observation can be partially explained by our argument in a). Another sensible explanation is that the layers farther away from the top learned more generic features that are immediately transferable to a new task. Since those layers are already in good shape, one can imagine that their gradients contain very little discriminative power.
>
> c) Computational overhead. Similar to fine-tuning, our method becomes increasingly expensive (in terms of both time and space complexity) as we include gradient from more layers. Hence, it helps to restrict ourselves to the topmost layer in order to reduce the computational cost. This choice is further backed by our arguments in a) and b).
>
> We apologize for not sticking with our claim in Sec. 4.1 in some of the succeeding experiments. We have performed additional experiments in which we use gradient only from the topmost layer (or residual block). Below please find a sample of the results.
>
> base network: encoder of deep generative models pre-trained on CIFAR-10
> target task: CIFAR-10 classification (top-1 accuracy)
> activation alone: 62.87 (BiGAN) / 52.05 (VAE)
> gradient alone: 70.14 / 61.47
> activation + gradient: 70.51 / 61.16
> fine-tuning: 71.78 / 65.16
>
> base network: ResNet-50 pre-trained on the Jigsaw pretext task
> target task: VOC07 object classification (mean average precision (mAP))
> activation: 57.83
> gradient alone: 57.73
> activation + gradient: 61.70
> fine-tuning: 67.88
>
> base network: ImageNet pre-trained ResNet-18
> target task: VOC07 object classification (mAP)
> activation alone: 82.65 (center crop) / 83.59 (ten random crops)
> gradient alone: 83.05 / 84.63
> activation + gradient: 83.50 / 84.95
> fine-tuning: 82.97 / 84.14

---

> > ### Author Response · Authors · 2019-11-15
> > **Response to Reviewer #2 Part II**
> >
> > [On extra ablation experiments]
> >
> > We first add an ablation study using an ImageNet pre-trained ResNet-18. The results show that our approach remains effective. More importantly, our model again outperforms fine-tuning, suggesting that the encouraging result on AlexNet is not observed by chance.
> >
> > We further consider the gradient term alone as another baseline in the two ablation studies. Interestingly, we found that this model is more powerful than the standard logistic regressor on network activation when all network parameters are pre-trained. Moreover, we compare our model against this same baseline in other experimental settings, and find that the full model generally outperforms the gradient model as the dataset and the network grows in complexity. This is highly desired as it makes our method potentially useful in practice.
> >
> > We present some key results of our ablation study in the supervised setting. Here, gradient feature is w.r.t. the last residual block of an ImageNet pre-trained ResNet-18. We form different combinations of random and pre-trained parameters in the base network for computing the gradient feature, while the activation feature is from the pre-trained network in all scenarios. Please see the appendix of the paper for detailed experimental design and the full set of results.
> >
> > random $\theta_1, \theta_2, \omega$: 15.63 (gradient alone) / 82.84 (full)
> > pre-trained $\theta_1$, random $\theta_2, \omega$: 62.35 / 82.84
> > pre-trained $\theta_1, \theta_2$, random $\omega$: 80.74 / 83.15
> > pretrained $\theta_1, \theta_2, \omega$: 83.05 / 83.50
> > activation alone: 82.65
> > fine-tuning: 82.97
> >
> > [On the difference between our work and the two mentioned by the reviewer]
> >
> > As can be seen from our general clarification, our work differs significantly from standard final layer transferring approaches, which only use network activation as feature. In fact, we consider it as a baseline in our experiments. Our method, on the other hand, uses additional feature, namely the Jacobian w.r.t. conv layers prior to the fc layer. Similarly, it is easy to distinguish our method from the Bayesian final layer approaches, which again only tune the fc layer.
> >
> > [On the implementation of our algorithm]
> >
> > Our implementation does not require sophisticated software engineering and can be trivially integrated into existing pipelines. Let us consider the toy example above and further assume that $\theta_2$ contains conv3 and conv4. Without loss of generality, we assume ReLU nonlinearity. The PyTorch pseudo-code goes as follows.
> >
> > v3 = Conv2d()			# v3 is the counterpart of conv3 in the second term
> > v4 = Conv2d()			# v4 is the counterpart of conv4 in the second term
> >
> > # standard forward prop (x is the image input)
> > y1 = relu(conv1(x))
> > y2 = relu(conv2(y1))
> > y3 = relu(conv3(y2))
> > y4 = relu(conv4(y3))
> > z = avgpool2d(y4)
> > out = fc(z)
> >
> > # code for scalable evaluation of the second term in our model
> > jvp = v3(y2) * (y3 > 0).float()
> > jvp = (conv3(jvp, add_bias=False) + v4(y3)) * (y4 > 0).float()
> > jvp = avgpool2d(jvp)
> > jvp = fc_copy(jvp, add_bias=False)		# fc_copy is a frozen copy of fc at initialization
> >
> > logits = out + jvp	  # logits are fed into a softmax for classification
> >
> > In transfer learning, conv1-4 and fc_copy are kept frozen. fc is the trainable parameter in the first term of our model, and v3, v4 are the trainable parameters in the second term. As the reviewer correctly pointed out, our method does introduce some extra memory overhead because some intermediate output needs to be cached (y3 and y4 in the example). However, it suffices to use gradient from conv4 in practice as we argued previously, hence the memory cost and the number of operations are on par with fine-tuning.
> >
> > [Typos]
> >
> > Thank you for pointing out the typos. They will be fixed in our updated draft.
> >
> > Reference:
> > [1] Wide Neural Networks of Any Depth Evolve as Linear Models Under Gradient Descent. NeurIPS 19

---

### Official Review · AnonReviewer1 · 2019-10-26
**Official Blind Review #1**

**Rating:** 3

**Review:**

Summary: This paper considers the use of a neural network's Jacobian as additional features for semi-supervised, unsupervised, and transfer learning of representations. The idea is simple and the authors motivate this choice by connecting it to the literature on the neural tangent kernel (although nothing is proven in this paper). Some experiments are performed in which the authors demonstrate some improvements over the simple baseline of using the neural network's final layer alone.

Strengths:
- The idea is simple and motivated by the Fisher vector work (Jaakkola & Hausler, 1999), which the authors cite.

Weaknesses:
- Comparing to a baseline with half as many features (just using the final layer of the neural network) is not a compelling baseline. The authors could have considered other alternatives: just using the gradients when comparing against the current baseline, augmenting the baseline with random features of the data.
- The discussion of the connection to theory added very little to the paper. This is not a theoretical paper, and the hand-wavy connections to the theoretical literature did not bolster the case. Instead, I found it distracting. I agree with the authors that this connection is important to point out, but a paragraph or two suffices.
- The ablation study could be more compelling. I'm not particularly surprised that a model improves as one increases the number of parameters that are pre-trained on the same kind of data (even if the pre-trianing objective is somewhat distinct).

**Experience Assessment:**

I have read many papers in this area.

**Review Assessment: Checking Correctness Of Derivations And Theory:**

I did not assess the derivations or theory.

**Review Assessment: Checking Correctness Of Experiments:**

I assessed the sensibility of the experiments.

**Review Assessment: Thoroughness In Paper Reading:**

I read the paper at least twice and used my best judgement in assessing the paper.

---

> ### Author Response · Authors · 2019-11-15
> **Response to Reviewer #1**
>
> Thanks for your valuable comments. We discuss additional results of alternative baselines that uses gradient feature alone. These results clearly show the advantage of our method, especially on challenging datasets and large-scale base networks. Moreover, results from our ablation studies rule out the explanation that the performance gain is due to the extra number of parameters. We hope that our response can address your concerns and kindly ask the reviewer to re-consider the rating.
>
> We present the key results in our response and put the full set of results temporarily in the appendix of the paper. These results will be merged into a future version of the paper. Please do not hesitate to ask if you have further questions.
>
> [On the alternative baselines]
>
> -- Augmenting the baseline with random features of the data.
>
> We indeed have considered this setting in the ablation study of our submission, with the results in the first row of Table 1. Specifically, we augment the activation from a pre-trained network (a BiGAN encoder trained on CIFAR-10) with the gradient from a random network, and the model achieves an accuracy around 63% for all three sizes of $\theta_2$. This is on par with the logistic regressor baseline, which achieves an accuracy of 63.38% by using activation alone from the pre-trained network. We thus conclude that our model's improvement over the standard logistic regressor is NOT a consequence of an increase in the number of parameters. We will revise the description of the experiment to better explain these results.
>
> --Just using the gradient as feature.
>
> We agree that using the gradient alone is an important baseline and thank the reviewer for pointing it out. This new baseline is referred to as the gradient model so that it can be distinguished from the full model. Surprisingly, the gradient model achieves impressive results in the case of toy datasets and small networks, sometimes even slightly outperforming the full model. However, the full model consistently outperforms the gradient model in the self-supervised and the transfer learning settings, where large networks (e.g. ResNets) and more challenging datasets (VOC07 and COCO2014) are used. A sample of our results that covers the most interesting scenario (i.e. when all network  parameters are pre-trained) is listed below. Please see the appendix of the paper full the full set of results.
>
> base network: encoder of deep generative models pre-trained on CIFAR-10
> target task: CIFAR-10 classification (top-1 accuracy)
> activation alone: 62.87 (BiGAN) / 52.05 (VAE)
> gradient alone: 70.14 / 61.47
> activation + gradient: 70.51 / 61.16
> fine-tuning: 71.78 / 65.16
>
> base network: ResNet-50 pre-trained on the Jigsaw pretext task
> target task: VOC07 object classification (mean average precision (mAP))
> activation alone: 57.83 (ten random crops)
> gradient alone: 57.73
> activation + gradient: 61.70
> fine-tuning: 67.88
>
> base network: ImageNet pre-trained ResNet-18
> target task: VOC07 object classification (mAP)
> activation alone: 82.65 (center crop) / 83.59 (ten random crops)
> gradient alone: 83.05 / 84.63
> activation + gradient: 83.50 / 84.95
> fine-tuning: 84.14
>
> Hence, We conjecture that our method can be more successful as the dataset and the network grow in complexity. This is highly desired as it makes our method potentially useful in practice. Finally, we have to point out that we find using gradient only is less intuitive and hard to interpret in its own right. Instead, the same gradient term acts nicely as a residual term derived from Taylor expansion in our model.
>
> [On the ablation study]
>
> As mentioned above, it is evident from our ablation study that our model improves on the baseline not just because it includes more parameters. At the request of Reviewer 2, we include another set of ablation study using a supervised network. The results from this new study further strengthens our conclusion. Please find the new results in the appendix of the paper.
>
> [On the discussion of the theoretical connection]
>
> We will follow the reviewer's suggestion to revise this section so that it can fit into two paragraphs. This change will be reflected in a future version of the paper.

---

### Official Review · AnonReviewer3 · 2019-10-29
**Official Blind Review #3**

**Rating:** 8

**Review:**

The key idea in this paper is to generate a feature vector based on a Fisher information idea for intermediary levels of a deep neural network. I liked the idea of using some form of linearization of the Fisher information matrix to learn a feature vector. The connection to tangent plane ideas in terms of robustness also made sense. The issue I have is what layer to apply this idea to and formally relating this idea to the variational VAE type framework. Using the top layer seems arbitrary. Also using one layer seems arbitrary. Is there a way to argue via a perturbation analysis of the variational problem of what makes the most sense. I feel the paper hints at this but does not make this idea explicit.

**Experience Assessment:**

I have published one or two papers in this area.

**Review Assessment: Checking Correctness Of Derivations And Theory:**

I carefully checked the derivations and theory.

**Review Assessment: Checking Correctness Of Experiments:**

I assessed the sensibility of the experiments.

**Review Assessment: Thoroughness In Paper Reading:**

I read the paper at least twice and used my best judgement in assessing the paper.

---

> ### Author Response · Authors · 2019-11-15
> **Response to Reviewer #3**
>
> Thank you for your valuable comments.
>
> We would like to address your concern on the selection of layers for computing gradient feature. In the experiments, we choose to obtain gradient feature from the topmost conv layers (or residual blocks) of the network for the following reasons.
>
> a) Width requirement. Empirical studies demonstrated that a network is well captured by its linearization only if it is sufficiently wide [1]. The most popular network architectures these days are almost always widest in the top.
>
> b) Diminishing gain. According to our ablation studies, the model using gradient from the topmost layer alone is already satisfying. Adding gradient from layers further down the hierarchy results in very little performance gain. This observation can be partially explained by our argument in a). Another sensible explanation is that the layers farther away from the top learned more generic features that are immediately transferable to a new task. Since those layers are already in good shape, one can imagine that their gradient contain very little discriminative power.
>
> c) Computational overhead. Similar to fine-tuning, our method becomes increasingly expensive (in terms of both time and space complexity) as we include gradient from more layers. Hence, it helps to restrict ourselves to the topmost layers in order to reduce the computational cost. This decision is further backed by our arguments in a) and b).
>
> Nevertheless, we admit that our current approach to layer selection is more or less heuristic, and believe that a more principled strategy must exist. Perturbation analysis sounds like a very interesting idea along this line, and we will definitely look into it in our future work.
>
> Reference:
> [1] Wide Neural Networks of Any Depth Evolve as Linear Models Under Gradient Descent. NeurIPS 19

---

### Decision · Program_Chairs · 2019-12-19

**Decision:**

Accept (Poster)

**Comment:**

The paper makes a reasonable contribution to extracting useful features from a pre-trained neural network.  The approach is conceptually simple and sufficient evidence is provided of its effectiveness.  In addition to the connection to tangent kernels there also appears to be a relationship to holographic feature representations of deep networks.  The authors did do a reasonable job of providing additional ablation studies, but the paper would be improved if a clearer study were added to investigate applying the technique to different layers.  All of the reviewer comments appear worthwhile, but AnonReviewer2 in particular provides important guidance for improving the paper.